# The Lower Paleolithic Engravings of Bilzingsleben, Germany

**Robert G. Bednarik** 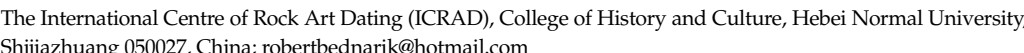

The International Centre of Rock Art Dating (ICRAD), College of History and Culture, Hebei Normal University, Shijiazhuang 050027, China; robertbednarik@hotmail.com

**Definition:** Some of the earliest known engravings are described, analyzed, and interpreted, following their microscopic examination. They are of significance in exploring the cognitive evolution of hominins several hundred thousand years ago and have not been described together before. The Steinrinne site near Bilzingsleben, north of Weimar, Germany, is one of Europe's most important Lower Paleolithic occupation sites. Its extensive human habitation floor, excavated over 1000 square meters, comprises some of the world's oldest evidence of dwellings, broadly matching or exceeding the age of examples proposed in Africa, India, and France. It has yielded numerous hominin remains, many wooden artefacts, other exquisitely preserved organic remains, and more portable engravings than any other Middle Pleistocene site. The latter are reviewed here, presenting the results of a detailed microscopic examination of the main finds. Bilzingsleben has so far produced the largest number of engraved Lower Paleolithic objects reported, which are particularly important to exploring the cognitive developments of hominins.

**Keywords:** earliest engravings; cognition; human evolution; Lower Paleolithic; *Homo erectus*

## 1. Introduction

If we are interested in the cognitive evolution of early humans and how they managed to create their first constructs of reality, the most important archaeological evidence we can expect to uncover consists of the finds considered art-like (i.e., paleoart). Such artifacts extend back to the Lower Paleolithic (roughly 2.6 million years to 250,000 years ago), spanning the earliest human stone tool use. The late part of that period yielded several forms of paleoart, e.g., beads, proto-figurines, engraved marks, or ochre use, the most common being surface engravings on various materials. The Bilzingsleben site has so far provided the largest number of such very early engravings [1–3], and they are reviewed here.

As they were published only in German initially [4], most English-speaking commentators remained unaware of the engraved finds from the Bilzingsleben Steinrinne until they were subjected to international debate in English [5]. Subsequent commentators claimed that these objects (and various others) were only published after their assessment of early symbolism [6] had appeared. This is a common problem for anglophone archaeologists; they are often unaware of material available in other languages [7]. One commentator initially accepted the authenticity of the Bilzingsleben markings [8], instead questioning their purported age [9]. After visiting the find site and examining the material, he conceded that he 'was wrong to give the impression that [he] could attack the stratigraphic dating to the Holstein interglacial complex' [10], but now he attacked the significance of the markings instead. Having earlier accepted their intentionality, based on superb photographs, he now argued that they were incidental cut marks.

Only four marked objects from Bilzingsleben have been described in detail in English. The organic residues from the Steinrinne site are exceptionally well preserved due to travertine precipitation from a mineral spring at the locality [11] and include large quantities of vegetational [12] and malacological [13] remains. Possibly about 350,000 years old [14], the site dates from the Holstein Interglacial (MIS 9e). The excavated 1000-square-metre

occupation floor shows distinctive patterning of activity zones [15,16], including apparent dwelling remains [17,18]. Lower Paleolithic windbreaks or dwelling foundations of similar or even earlier ages have been reported from various sites in Eurasia and Africa [19]. Twenty-five cranial fragments and seven molars from the site are said to be of a late *Homo erectus* [20], whose well over 100,000 recovered artefacts include polished ivory points, wooden staffs, and a series of incised objects. The animal remains feature a high proportion of large mammals, especially rhinoceros (26.6% of mammalian individuals), with extensive evidence of systematic butchering. The marked objects also consist primarily of bones from large animals [5].

Engraved object No. 1, made from the flat spall of the tibia of a straight-tusked elephant (*Palaeoloxodon antiquus*), was probably used as a percussion tool. Object 2 from Bilzingsleben is the distal part of a flat rib bearing a series of identically angled parallel incisions over the width of the rib. Each mark consists of several separate applications of the engraving tool, which suggests a careful, intentional procedure. The object is unsuitable as a cutting board, and no convincing utilitarian explanation for the marks or their consistent arrangement has been offered. Object 3, a retouched artefact believed to have been used for woodworking, is again from an elephantine bone. It bears a series of long convergent lines of extraordinary straightness. They are evenly engraved, and the multiple applications of a particular tool point with two minute projections is evident. Object 4 is a flat piece of bone with a series of more cut marks, all of which are thought to have been made by the same stone implement.

In addition to these four previously described artefacts, four other engraved objects have been excavated at the Bilzingsleben hominin site. They were found among significant numbers of bone and antler fragments bearing taphonomic (e.g., trampling) or defleshing marks and include four items to which we wish to draw attention. First, another bone artefact should be considered here, bearing numerous linear markings forming a consistent pattern. A large polished ivory point made from a split elephant tusk also warrants attention. The site has yielded numerous examples of large bone and ivory artefacts that were expertly split with wedges, involving a complicated manufacturing process. An apparently non-utilitarian, intentional marking occurs also on a slab of quartzite. Finally, another bone bearing a set of convergent lines has only been found and reported recently.

The site's location is N 51°16′20.27″–E 11°03′34.87″. Its elevation is 168 m a.s.l., and it is situated about 1.2 km southwest of Bilzingsleben township and accessible from there by road.

## 2. Bilzingsleben Object No. 1

*Object Description*

This flat spall from an elephant tibia is 395 mm long, 120 mm wide, and 65 mm thick (Figure 1a–c). The extensive fracture seen at one end and some impact marks along an opposing edge suggest that it was used as a cudgel or club. The fracture has truncated a set of 14 remaining engraved straight lines fanning out along the narrow bevel that forms the spall's long side. The lines were incised with a stone tool. They are evenly cut and spaced, forming two structured complementary sets occupying most of the remaining bevel surface. Their narrow microscopic sections suggest they were all incised with the same tool point as single strokes. Some commentators have argued that these carefully made grooves may have been produced incidentally when the artefact served as a cutting board, for instance, to cut animal skin into strips. However, this interpretation overlooks that the flat, unmarked upper surface of the bone slab is much larger and well-suitable for this purpose. In contrast, the side bevel is virtually impossible to use effectively as a cutting support. A laser-microscopic study of this and the other Bilzingsleben engravings has determined that the markings were produced deliberately and with great care [21]. The evenly spaced and purposefully made markings are intentionally arranged in a distinctive pattern. They would be regarded as non-utilitarian without hesitation if found in an Upper Paleolithic context. However, the age of an object should not influence its classification.

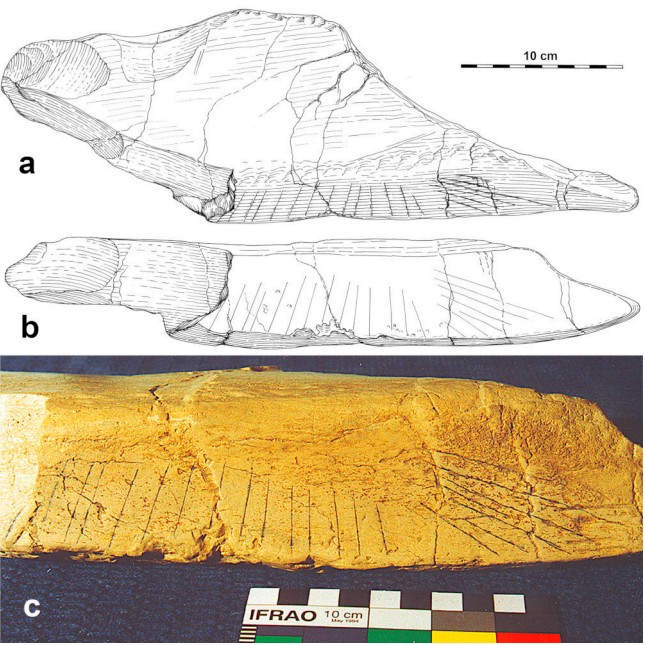

**Figure 1.** Three views of Bilzingsleben engraved object No. 1; (**a**,**b**) after Mania and Mania (1988) [5], (**c**) by author.

Although we cannot know how the sequence of engraved lines continued past the ancient fracture, we do know that there are two sets of convergent lines. Those of one set are longer than the other and inclined at a steeper angle. Amazingly, a similar engraved bone has been found only 10.5 km from Bilzingsleben, at Oldisleben. It also bears two distinct groups of linear cuts. However, this specimen is significantly more recent, in the order of one-third the age of the Bilzingsleben assemblage, having been recovered together with Eastern Micoquian or Keilmessertradition lithics. The lines are more deeply cut on the Oldisleben scapula fragment, having been made with multiple tool applications and up to 550 μm deep (Figure 2). Although the marking strategies differ between these two artefacts and are rather more developed on the younger specimen, it seems incredible that such a graphic convention could have been in use for such a significant time span.

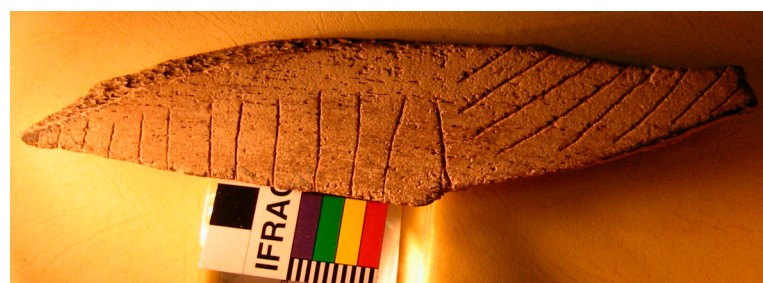

**Figure 2.** Engraved scapula fragment from Oldisleben, resembling the pattern on Bilzingsleben object No. 1; image by author.

## 3. Bilzingsleben Object No. 2

### 3.1. Object Description

The excavation number of this rib fragment is B1 33, B1 132; the Halle Museum number is 2002: 3033. This distal fragment of a flat but slightly curved rib of an undetermined large mammal displays a series of parallel, similarly angled markings incised with a stone tool on its outer surface. From the left in Figure 3, the first four irregularly spaced markings, each consisting of three or four partly overlapping cut marks. So, the tool used was raised each time and re-engaged, continuing in the same direction. Repeated in all four markings, this

careful procedure indicates deliberate action rather than thoughtless, automatic doodling. Further to the right is a single engraved line not extending over the entire width of the rib fragment. Near the fractured right end of the fragment, truncated by the fracture, is a double line, also engraved at the same angle. Morphologically, all these engravings are so similar in their section that it has been proposed that they have been made with a single tool (Figure 3a).

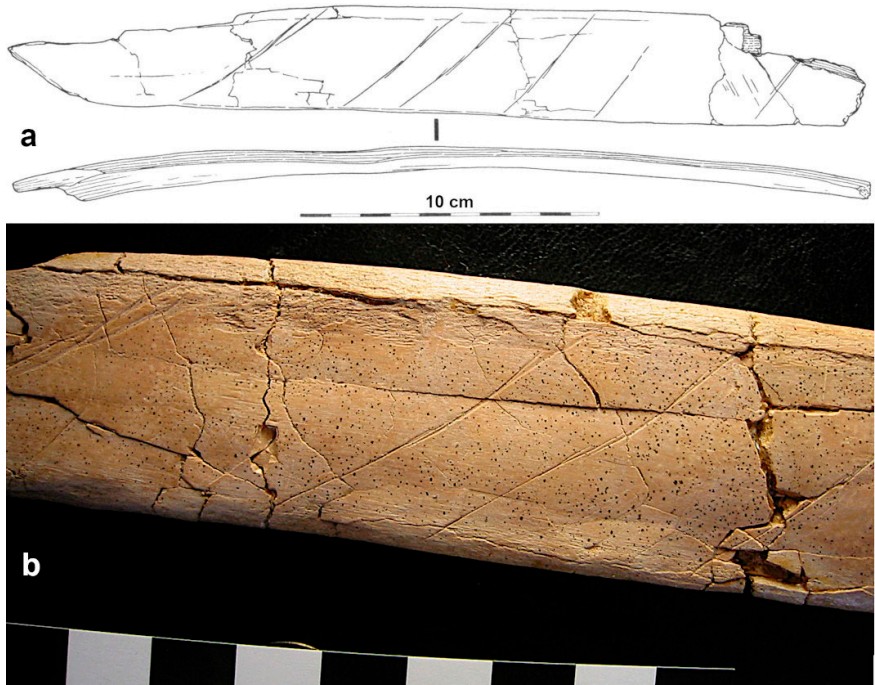

**Figure 3.** Three views of Bilzingsleben engraved object No. 2; (**a**) after Mania and Mania (1988) [5], (**b**) by author.

### 3.2. Markings

The incised grooves on one side of this elongated object are unusually deep, narrow and, for their age, exceptionally well preserved. Excellent striations occur in numerous places. Some groove sections are up to 160 microns deep, being of only about 240 microns width that soon narrows to under 150 microns. Apart from some isolated and perhaps incidental lines, there are several very distinctive sets of incisions, all occurring at about the same angle to the margins of the engraved area. Few marks occur in the intermediate spaces. These groove sets are drawn diagonally from the right upper to the left lower, and there is a clear intention of tight grouping that seems to exclude the possibility of interpretation as defleshing products. The sets of grooves tend to commence with a broadly abraded furrow comprising numerous striations, while the second tool application, through the convex middle part of the rib, is much deeper and narrower. These grooves were not made with stone points but with the cutting edge of a flint tool in a sawing rather than engraving action (Figure 4). This thin tool edge comprised a few minor asperities that left traces in many of the cuts. This tool action resulted in distinctly undercut groove sections in the rib's central part. Because of these characteristics, it would be difficult to confirm that the same tool was used in each individual mark or set of grooves because grooves tend to differ considerably according to the convexity of the bone surface, angle of application and relative position of the tool edge. However, some similarities, particularly in the deeply cut cross-sections, suggest that the same cutting edge is likely to have been used in at least some of the groove sets.

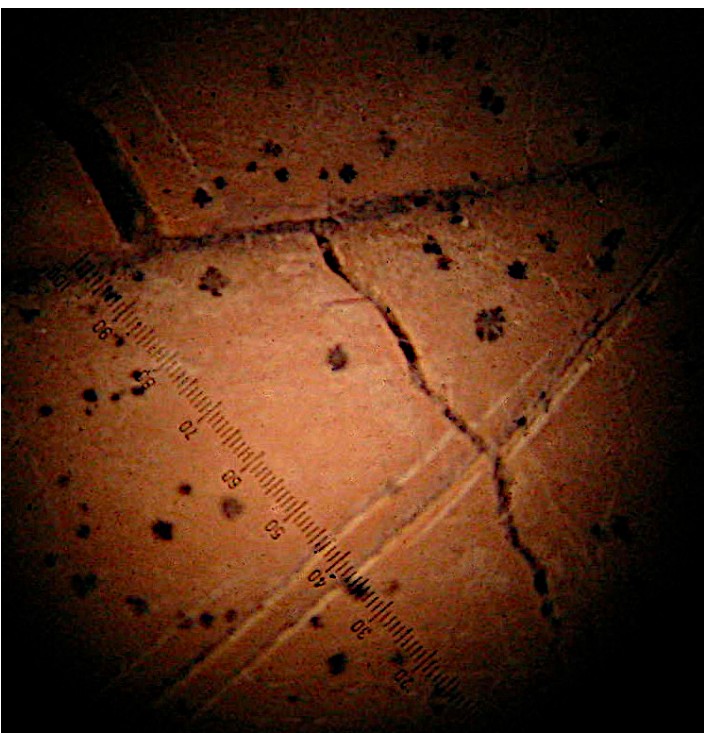

**Figure 4.** Microphotograph of engraved line on Bilzingsleben engraved object No. 2.

Beginning from the artefact's left (when viewed in the direction of marking production, assuming that marks were made from the top), the first set consists of four separate applications, presumably of the same cutting edge (Figure 3b). The most recent is the one nearest the upper edge, which peters out at the prominent crack. The second set has been made by three tool applications and is somewhat better pronounced. Here, the upper marking is particularly broad, up to more than 0.5 mm wide, bearing numerous striae. It is followed by a very narrow, deeply cut groove in which the tool edge had been held between 50 and 70 degrees to the bone surface rather than vertically. The third line, completing the set, is a single groove lacking striations.

The same marking behavior is evident in the third set from the left, but the procedure was with more precision here. The first marking is the more narrow; the second is again undercut at the same angle and very deep, but further down, it peters out. It is then precisely duplicated by the same tool in a parallel groove, but this line stops suddenly. It is followed by a fourth application of probably the same tool, endeavoring to continue the interrupted line in the same direction, running evenly to the edge.

The fourth set could be of a single tool application, but because of the curvature of the bone surface, the groove section changes considerably. In the central part, several asperities along the cutting edge of the stone implement come into play, where the groove reaches a maximal width of about 875 microns, with numerous striations. Further along, the groove narrows, and the damage in this area makes it unclear whether the tool application was continuous. A distinct offset against the line's upper part could indicate that the tool was applied a second time at this point. The groove then continues with a deep and narrow section, but faint parasitic traces again indicate that a tool edge rather than a tool point was applied.

Group five, near the other end of the artefact, consists of the traces of three tool applications that follow a pattern and angle similar to those of previous sets. The same tool edge could have well made them, and again, there is a degree of undercutting, indicating that the working edge was held at an inclined position relative to the bone surface.

There are a few isolated grooves between groups four and five, the longest of which has a distinctive starting mark showing the impression of a stone point. This differs from

the grouped and structured marks on this object and may have been made by a different tool or another aspect of the same tool.

### 3.3. Interpretation

The rib fragment bears engraved decoration that, technically, differs significantly from the other Bilzingsleben specimens. It was marked with a cutting implement rather than a point. However, the markings appear too precise to be seen as defleshing or taphonomic effects. The consistency in the angle, the deliberate repetition of the marking strategy and the very consistent tool application in each of the several sets imply intentionality. However, at the same time, it must be appreciated that the patterning or marking behavior evident in this item has not been reported before from this early period, and in that sense, the specimen remains unique.

## 4. Bilzingsleben Object No. 3

### 4.1. Object Description

The object's original find number is Bi 260,55, and the Halle number is 2002: 3032. It consists of a tibia fragment of the straight-tusked elephant (*Palaeoloxodon antiquus*), the fractured side of which is entirely flat (Figure 5a). It is 141 mm long, 61 mm wide, and 20 mm thick. One edge bears continuous retouching on the dorsal side, while the flat ventral side has given rise to a set of prominent engravings. These are perfectly straight grooves up to 70 mm long and arranged, forming a group converging on the pointed ends of the engraved surface.

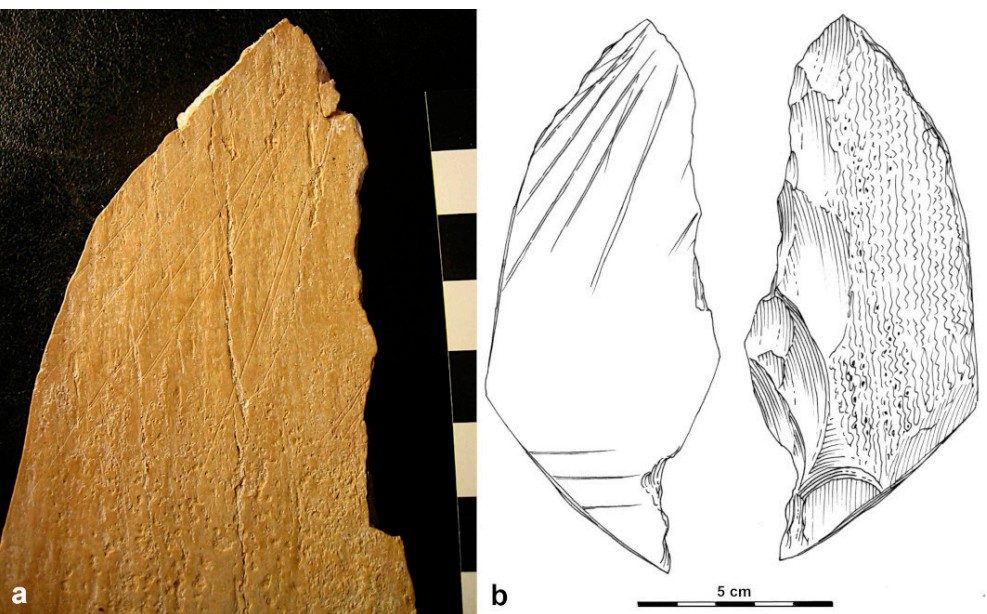

**Figure 5.** Three views of Bilzingsleben engraved object No. 3; (**a**) by author, (**b**) after Mania and Mania (1988) [5].

### 4.2. Markings

This object consists of a flat bone plaque with a spongy underside and an almost perfectly flat, smooth 'upper' panel bearing all markings. It was marked in its essentially present overall form, with most or all edge facets as they appear now. This is shown by the several instances of engraved lines that extend slightly onto side surfaces or at least run off the edges formed between them and the flat engraved panel. The importance of this observation is that lines were arranged in the available room as they are now, which facilitates consideration of their spatial arrangement relative to boundary and panel shape. Notably, the long lines, which all seem to emphasize the shape of one of the ends of the elongated plaque and thus converge towards a common focus, do not approach this point

but commence a short distance from it and thus from the margin of the available space. Microscopic examination of these long lines suggests they were all drawn from the point 'downwards'. The three short transverse lines near the other end of the object were drawn from the interior towards the edge, two of them relatively lightly. The central of these three lines runs distinctly over the margin, as do three of the long lines, which also run off the left edge. Apart from this feature, the direction of engraving is judged by details at minute rises or in rare superimpositions.

With one exception, the bundled or convergent long lines commence shallow and then deepen. Whenever they pass a rise or are deeply incised, those made with a specific two-pointed tool exhibit the 'parasitic' second line caused by the second point, spaced 300 to 400 microns from the first. The grooves are exceptionally well preserved, and longitudinal striations can frequently be observed in them. Only a few superimpositions of engraved lines occur, and fewer still provide microscopic features permitting determination of precedence. Nevertheless, the prominent short line that crosses over two of the long convergent lines, defying the overall pattern, can be seen to precede the double line it touches near its lower end. The isolated five short lines along the 'right hand' edge cannot be related to the other decorations chronologically or structurally, but they, too, clearly postdate the edge they touch.

The principal set of decorations, the long convergent lines, permit some assessment in that respect. The uppermost line that seems to have been made by the double-pointed tool (the duplicate line occurs only in one location, near its lower end) appears to postdate the longer line it crosses, but this is unclear. Above it is another finer line that crosses the first of the convergent lines, sub-parallel to the first mentioned, and postdates the long double line. Of the five long convergent lines, the third and fourth are intersected by a short incision that precedes them. Since the six lines made with the same double-pointed tool can be assumed to be contemporary and made in a single sequence, it follows that the short 'diagonal' groove was made first, with either the same or another tool, followed by the five long lines. Then, the sixth, shorter double line was added near the upper end. Finally, the uppermost fine line was superimposed, probably still with the same tool. Its relationship is not clear to the last of the long lines, the second from the right, which provides no indication that it was made by the double-pointed stone tool. It runs close to and sub-parallel to the lowest long line as if it were a duplication. Its groove section is distinctly broader. However, its course and alignment suggest that it may result from turning the tool point and that it was added to emphasize the lowest groove and the set as a whole. Unfortunately, there is no superimposition, and this question must remain open.

The five double lines are particularly interesting, which can even be recognised in the specimen's photographs. In all cases, the primary line is on the left and is narrow and smooth. The secondary line, where it is fully developed, is broader and shallower, and in each case, there are two vague striations representing two subsidiary points, about 120 μm apart. Where the groove section is best preserved, this secondary groove has an almost square profile at $80\times$ magnification, as if this point had been rather chisel-like. The repetition of this distinctive pattern in the five double lines provides conclusive evidence that, in each case, the same tool point was applied. It was similarly positioned relative to the direction of movement, i.e., the same groove profile was produced (Figure 6). Of interest, there is a series of 'drying cracks' along the object's center, and the lines crossing these gaps are distinctly offset by the relative displacements caused. Indeed, the engravings were made before these cracks formed.

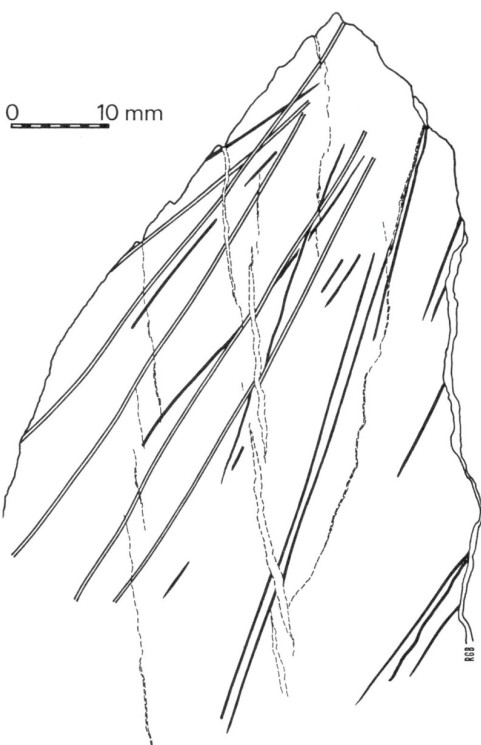

**Figure 6.** Analysis of the main body of engraved lines on Bilzingsleben object No. 3.

*4.3. Interpretation*

The line markings on this bone plaque are apparently all caused by stone tool points. One distinctive point profile has been securely identified, occurring in five lines that vaguely form a convergent group. This group is thought to have been made in a single sequence, preceded by one previous marking of a different orientation. The uppermost of the double lines was added after the second one below it had been executed, and a faint line was also superimposed. The chronological relationship of the remaining markings is not known. As far as can be established, all lines were drawn from the top, mostly not commencing from the margin of the panel. The arrangement of the main group and some subsidiary lines, possibly made with a different tool, is deliberate, emphasizing the panel shape near the top but avoiding the other lines. A marking strategy of 'filling the available area' is also evident in Bilzingsleben's specimen No. 1.

The best possible interpretation of the empirical observations is that the main group on this plaque is a structured set of unusually long, very straight, and deliberately incised lines made with one tool, without changing the grip on the tool, in one sequential action. The grooves were not made with cutting edges, as those on object No. 3; they were made with points and were not the by-product of a technological process (hide cutting). Their production was direct, not cutting through some intermediate material, and tool pressure was significantly even over much of each line. The grooves were drawn slowly, deliberately, and systematically, and the straightness of the lines is particularly notable.

## 5. Bilzingsleben Object No. 4

*5.1. Object Description*

The object's original find number is Bi 182,32, and its Halle Museum number is 2002: 3030. The flat bone fragment is 114 mm long, 55 mm wide and 9 mm thick. Its slightly convex surface bears a series of sub-parallel engraved lines, some evenly spaced.

*5.2. Markings*

All except one of the nine grooves forming the main set on this plaque are made by the same tool point, held and applied in the same way and direction. This is evident from the

remarkable consistency in the groove section. When viewed so that the lines were drawn from the top, each groove is deeper on the right than the left. The base of each groove is thus sloping but straight in each section, about 175 microns wide, and the angle of the groove floor is identical in all markings of the main set. The only exception is the first line on the left, which is of a different orientation and precedes the following line, as indicated by the latter's superimposition. One of the lines of the main set possesses a distinctive impact or pressure mark at its point of commencement, illustrating the forceful application of the tool point. The grooves mainly extend to the edge of the porous bone sponge area; two extend into this zone, thus postdating that surface.

In addition to this set of sub-parallel grooves, several markings might be taphonomic in origin, although some have been occasioned by stone tool asperities. In particular, two of the three grooves forming a set that now straddles the largest drying crack were made by a single tool point with a distinctive morphological signature of two striae. A few shorter lines to the left of the main set are exceptionally narrow (well under 160 microns) and may have been made by a single tool point. However, no compositional structuring is evident in these random markings (Figure 7).

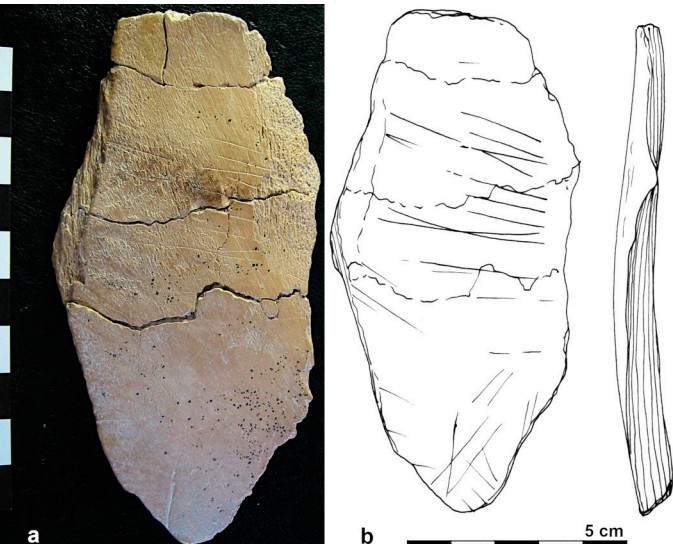

**Figure 7.** Three views of Bilzingsleben engraved object No. 4; (**a**) by author, (**b**) after Mania and Mania (1988) [5].

*5.3. Interpretation*

This is an excellent example of a set of sub-parallel engraved lines made by one tool point, held similarly with each application. It was created in a single sitting, and the deliberate application of the tool is also evident from the avoidance of the sponge zone, the mostly consistent spacing, and the execution as a discrete group that mirrors the marking strategies on objects 1 and 2. This specimen, too, is a bone plaque bearing intentional decoration.

## 6. Bilzingsleben Object No. 5

*6.1. Object Description*

The original find number is 81/B1; it had not yet received a Halle Museum number when examined. This elephant phalanx has a 172 mm maximum length, and is 106 mm wide and 52 mm thick. Except for marginal damage, this specimen is preserved completely. However, in contrast to the impeccable condition of preservation evident in the other engraved objects from Bilzingsleben, the piece is very corroded, and its numerous markings are less distinct. Nevertheless, the use of stone points is still evident in many of the grooves, so the specimen must be considered with the engraved objects. There are linear markings on the bone's convex and concave surfaces, but only those on the concave surface are

considered here. The markings on the strongly convex side of the object appear to be mainly of taphonomic origins of various types.

Some dozens of lines are present on the phalanx, made by stone points as indicated by parallel, faintly preserved striations in some instances. Collectively, they provide an impression of a rectangular structure with some diagonal lines, arranged to occupy the available space (Figure 8). The arrangement has been placed centrally on the panel, and repeated tool applications to lines are evident. These factors imply a considerable level of intentionality, and if this were correct, object No. 5 would bear the most complex of the Bilzingsleben engravings.

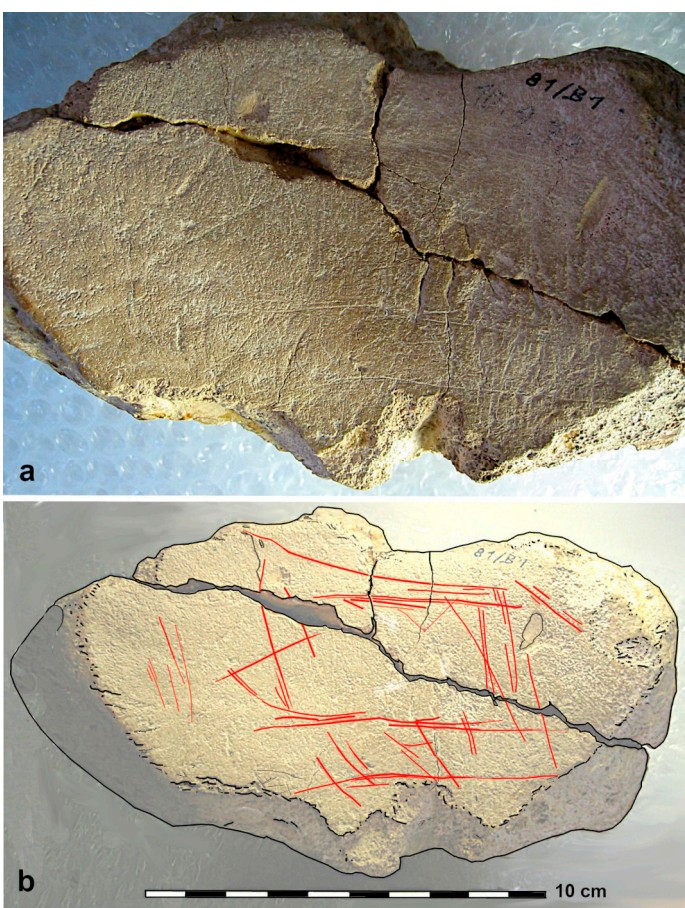

**Figure 8.** Bilzingsleben engraved object No. 5 (**a**); (**b**) indicates the engraved pattern in red lines. Both images are by the author.

### 6.2. Markings

Due to the poor preservation of most of the markings, it must be cautioned that many may be incidental or taphonomic. However, some aspects convey an impression of intentionality, such as the overall rectangular area of marking and its delimiting lines. Most notably, one long line roughly parallel to the edge of the panel appears to represent an attempt to form a very straight marking, clearly made with a stone point. This is emphasized by a shorter second line drawn parallel to it, about 14 mm away, also perpendicular to the predominating direction of several worn grooves. Another factor implying intentionality is the diagonal elements evident on this panel. Two deep and perfectly parallel grooves of 36 mm in length, 1.8 mm apart, are offset against the rectangular central area. Another incision crosses through the central part at roughly the same angle. A similar double line, which may result from a double-pointed graver, is again approximately parallel but much longer. Faintly curved, it runs from one margin to the other over a length of 74 mm.

The rectangular marked area is on one side delimited by an elongated depressed patch that seems to result from repeated abrasive tool application that has worn away much material. The overall arrangement is central on the bone panel, and despite the poor preservation, repeated tool applications to some of the best-preserved lines are evident.

*6.3. Interpretation*

The many dozens of lines on this phalanx could be explained away as forming geometric arrangements by pure coincidence. However, many were demonstrably made with stone points, as indicated by groove sections, parallelism, and occasionally even very faintly preserved striations. The markings provide an impression of a structure (rectangular, with distinctive diagonals at angles of roughly 45 degrees) arranged to fit into the available space, the concave surface. Notably, several of the more prominent markings appear to connect with the ends of others at either 90° or 45° angles. Based on probability, it is difficult to see how these features could all be coincidental, even though that possibility may not be soundly excluded.

Considering that several items from the Bilzingsleben deposit feature intentionally engraved markings, the probability that the poorly preserved No. 5 object bears a purely fortuitous arrangement becomes even less persuasive. The main argument against the intentionality of its markings is their very complexity. If they formed an intentional arrangement, it could be the most sophisticated Lower Paleolithic engraved patterns currently available. This, however, should not be a legitimate argument against its acceptance. Several features appear (in terms of 'graphic behavior' traces) that are reminiscent of the much more recent Blombos Cave hematite pieces with geometric markings and other Pleistocene engraved markings.

A complex arrangement has been incised on the concave surface of a complete metatarsal bone, again of *Palaeoloxodon antiquus*. Neither the structure of the marking nor its relationship to its support suggest a utilitarian origin. The bone is hardly suitable as a cutting board, and no alternative explanation has been offered for the marks. It is, of course, true that any random arrangement of superimposed cut marks will result in some kind of geometric pattern. However, in the case of this arrangement, it seems far-fetched to explain its geometry as simply fortuitous. There is an apparent relationship between the spatial distribution of the incisions and the borders of the available area, which random markings on a concave surface would not be expected to reflect, and there are several marking strategies that indicate intentionality. If this arrangement was made intentionally—which does seem likely—then it indicates a significantly more advanced level of concept-mediated marking behavior than has been attributed to these hominins in the past. Most certainly, object No. 5 from Bilzingsleben warrants much more detailed investigation and precision recording of the arrangement it bears, despite its poor preservation.

## 7. Bilzingsleben Objects Nos 6 to 8

*Object Descriptions*

No. 6. The 65 cm long ivory point is incompletely preserved and was perhaps used as a thrusting or piercing weapon or as a lance point on which charging large game would impale itself. The size suggests that it may have been attached to a wooden staff. It bears two well-executed parallel arcs of about 30 mm diameter, engraved into the polished splicing surface of this remarkable ivory artefact (Figure 9a), the earliest known of its kind [22]. There is no plausible explanation for why the prominent marking might be fortuitous. Both marks show identical evidence that, in each case, the tool point slipped in creating the technologically challenging curvature.

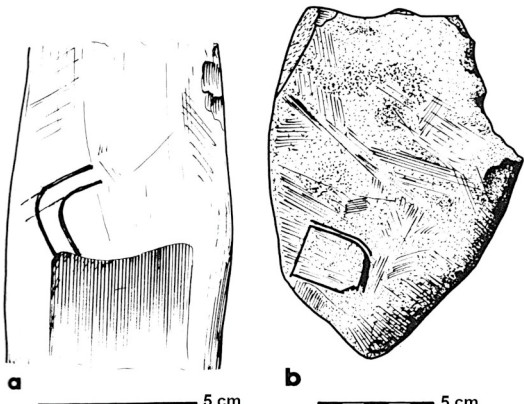

**Figure 9.** (**a**) Double arc cut into a polished ivory point; (**b**) D-shaped engraving on a quartzite slab; both from Bilzingsleben. Both images are by the author, previously published [23].

No. 7. Item No. 177,17 is a quartzite slab, 15 cm long. The marking is a well-executed, elongated D-shape, 35 mm long and 25 mm wide. Two aspects are of particular interest. The marking was engraved with several strokes, and the artisan seems to have experienced difficulties in symmetrically shaping the curved part of the figure, correcting the line several times in the process (Figure 9b). This treatment indicates, even more than in the above cases, the full intentionality of the process. Moreover, the particularly tough nature of the support renders implausible the assumption that the slab may have been a cutting board: no Paleolithic person would foolishly damage a stone tool by using a quartzite support. Most of all, it needs to be appreciated that the creation of this marking involved a great effort and a significant challenge to Lower Paleolithic tools—as any replication attempt would demonstrate.

No. 8. This object was only discovered in 2014 and introduced to the discipline four years later, in Session 3 of the NeanderART 2018 conference held at the University of Turin from 22 to 26 August 2018. It was presented by Enrico Brühl, the Scientific Director of the Archaeological Museum 'Steinrinne' Bilzingsleben [24]. The object is a frontal fragment of a cervid metacarpal bone that bears five deliberately arranged cutmarks that form a fan-like pattern reminiscent of the divergent lines array found on object No. 3, perhaps even capturing the idea expressed in No. 1's design. Again, the marks are evenly spaced and arranged symmetrically. The three central lines, made in single strokes, seem to have been of similar lengths but were truncated by bioturbation damage. The two lateral marks are 5.5 and 7 mm long, each made by two cuts, like those on object No. 2 (Figure 10d). Laser-microscopic scans suggest that the markings are all uniform in depth, width, and profile and have all been made using the same stone tool. The pattern on object No. 8 shows that the convergent lines motif, so prominent in the earliest known palaeoart of the world, occurs several times at Bilzingsleben.

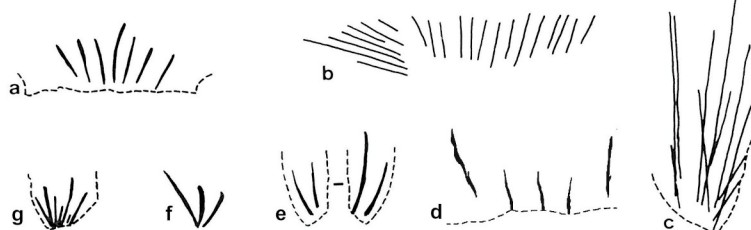

**Figure 10.** Engraved convergent lines motifs of the Lower (**a–d**) and Middle Paleolithic (**e–g**), from Stránská Skála 1 (**a**), Bilzingsleben (**b–d**), and Prolom 2 (**e–g**). Drawings not to scale, image modified from Bednarik (1995) [23].

## 8. Discussion

Indeed, the engravings on Bilzingsleben object No. 8 closely resemble those on a juvenile forest elephant vertebral fragment from the Czech site Stránská Skála 1, except the latter specimen features seven short convergent lines, rather than five [23,25]. Stránská Skála was occupied about the same time as Bilzingsleben and reportedly also by late *Homo erectus*. We have already mentioned the similarities between Bilzingsleben's object No. 1 and one of the much younger engraved bone fragments from Oldisleben, attributed to the Eastern Gravettian. That Middle Paleolithic tool tradition, also known as the Keilmesser industry, has yielded another three objects bearing convergent lines motifs at Prolom 2, Crimean Peninsula, Ukraine [23,26]).

The principal challenge provided by those opposed to pre-Upper Paleolithic engravings or other paleoart forms is to posit that such markings provide no standardized conventions of Lower and Middle Paleolithic designs, i.e., there are no repeated patterns of engravings. However, finds from Bilzingsleben, Stránská Skála, Prolom, and Oldisleben invalidate this opinion. The convergent or fan-like markings engraved on eight bone objects are either aligned with a 'dominant' edge of the object, thus emphasizing it, or they accentuate a pointed aspect of the object's form (Figure 10). In all cases, they are spatial responses to the area available for decoration, and many of them are symmetrically placed or arranged. If we see this in the context that some of the early engravings would have been made with significant difficulties (especially the Bilzingsleben No. 7 engraving on quartzite or the four deep grooves on the Prolom horse canine; Figure 10e), we can appreciate the determination of the makers. These factors demonstrate deliberate action to produce markings complying with standardized conventions that are no less explicit than those of Upper Paleolithic 'art'. It is apparent that the convergent lines motif is one of the most common of Modes 1, 2, and 3 lithic industries, exceeded in number only by cupules and sets of parallel lines.

**Funding:** This research received no external funding.

**Institutional Review Board Statement:** Not applicable.

**Informed Consent Statement:** Not applicable.

**Data Availability Statement:** Further data are available from the author.

**Conflicts of Interest:** The author declares no conflict of interest.

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
