# Peer review of "The Lower Paleolithic Engravings of Bilzingsleben, Germany"

_encyclopedia, doi:10.3390/encyclopedia4020043_

Round 1
Reviewer 1 Report
Comments and Suggestions for Authors
Dear Authors
You can access my comments and suggestions regarding your valuable data through the PDF attachment related to your manuscript.

Author Response
Reviewer 1 has raised 7 comments in the paper's PDF. All of them relate to the suggestion that there should be more interpretation of the material from Bilzingsleben and some level of speculation about intent. As this is submitted as an encyclopedia entry, which is supposed to present only soundly established empirical data about the subject, I cannot comply with these requests without compromising the integrity of the paper.
Reviewer 2 Report
Comments and Suggestions for Authors
The Lower Paleolithic engravings of Bilzingsleben, Germany
This is a worthy contribution to the study of the first symbolic expression of human groups from the excellent archaeological record register at the Bilzingsleben archaeological site. This place has provided some of the most relevant pieces of the Lower Paleolithic record.
This study pays attention to a relevant sample of pieces using an analytical description and a later microscopical review. A discussion of the marks and the distribution provides an interesting framework for its interpretation.
From my point of view, this study deserves to be published with minor changes due to the orientation of it and the importance of the studied material. However, the traditional approach to these pieces lacks an important aspect, which is the use of ethnographic and experimental comparative material. In this direction, the use of these sources could provide important material for the interpretation of the distribution and the morphology of the traces. Perhaps, and inclusion of these aspects in the conclusions, for future approaches would enrich this excellent paper.
Author Response
This "excellent paper" does not include ethnographic interpretation or experimental work of speculative interpretation. It is intended as an encyclopedia entry, which is meant to provide only solid empirical data. Referee 2 offers no other criticisms.
Reviewer 3 Report
Comments and Suggestions for Authors
Dear author
The article elaborates on the revelation of some of the earliest engravings discovered at the Steinrinne site, located near Bilzingsleben, north of Weimar, Germany. These engravings have been meticulously examined through microscopic examination, providing insights into the cognitive evolution of hominids several millennia ago. Predominantly documented in large faunal remains, these Lower Paleolithic engravings provide a platform for exploring the cognitive progression of our ancient ancestors. Although the structure of the article is sound, one suggestion for the author would be to expand the discussion section, as it leaves readers longing for a more comprehensive discourse on the results obtained, where he could present and contrast hypotheses on the cognitive development of the hominids responsible for these engravings.
Author Response
The supportive review is well appreciated, but the one comment offered cannot be satisfied. This is intended as an encyclopedia entry, which should only include soundly established empirical information. The discussion section is as far as I am able to go in terms of interpretation based on what is soundly known.
Reviewer 4 Report
Comments and Suggestions for Authors
The article seems interesting and publishable in its current format, without any changes. The author is a renowned specialist in the field, and it concerns a set of pieces of great interest. Overall, it addresses a relevant and interesting topic showing the bones with marks from Bilzingsleben. Regardless of this reviewer's opinion, as I believe there are still many more potential analyses that could be made on these remains to validate many of the conclusions about the different possible engravings, the article is of great interest and utility to researchers interested in this subject.
Author Response
Reviewer 4 does not require any changes to the paper and states that it can be published "in its current format". The reviewer recognises that there are many more potential analyses after the basic empirical data of the paper have been made available. That is the entire purpose of the paper, and I thank the reviewer for recognising this need for a sound empirical basis.